# A Ropy Exopolysaccharide-Producing Strain *Bifidobacterium pseudocatenulatum* Bi-OTA128 Alleviates Dextran Sulfate Sodium-Induced Colitis in Mice

**DOI:** 10.3390/nu15234993

**Published:** 2023-12-01

**Authors:** Hui Wang, Xinyuan Zhang, Xinfang Kou, Zhengyuan Zhai, Yanling Hao

**Affiliations:** 1College of Food Science and Nutritional Engineering, China Agricultural University, Beijing 100083, China; 2Key Laboratory of Precision Nutrition and Food Quality, Department of Nutrition and Health, China Agricultural University, Beijing 100083, China; 3Food Laboratory of Zhongyuan, Luohe 462300, China

**Keywords:** *Bifidobacterium pseudocatenulatum*, colitis, exopolysaccharide, intestinal barrier, gut microbiota

## Abstract

Inflammatory bowel disease (IBD) is a chronic disease associated with overactive inflammation and gut dysbiosis. Owing to the beneficial effects of bifidobacteria on IBD treatment, this study aimed to investigate the anti-inflammation effects of an exopolysaccharide (EPS)-producing strain *Bifidobacterium pseudocatenulatum* Bi-OTA128 through a dextran sulfate sodium (DSS)-induced colitis mice model. *B. pseudocatenulatum* treatment improved DSS-induced colitis symptoms and maintained intestinal barrier integrity by up-regulating MUC2 and tight junctions’ expression. The oxidative stress was reduced after *B. pseudocatenulatum* treatment by increasing the antioxidant enzymes of SOD, CAT, and GSH-Px in colon tissues. Moreover, the overactive inflammatory responses were also inhibited by decreasing the pro-inflammatory cytokines of TNF-α, IL-1β, and IL-6, but increasing the anti-inflammatory cytokine of IL-10. The EPS-producing strain Bi-OTA128 showed better effects than that of a non-EPS-producing stain BLYR01-7 in modulating DSS-induced gut dysbiosis. The Bi-OTA128 treatment increased the relative abundance of beneficial bacteria *Bifidobacterium* and decreased the maleficent bacteria *Escherichia-Shigella*, *Enterorhabuds*, *Enterobacter,* and *Osillibacter* associated with intestinal inflammation. Notably, the genera *Clostridium sensu stricto* were only enriched in Bi-OTA128-treated mice, which could degrade polysaccharides to produce acetic acid and butyrate in the gut. This finding demonstrated a cross-feeding effect induced by the EPS-producing strain in gut microbiota. Collectively, these results highlighted the anti-inflammatory effects of the EPS-producing strain *B. pseudocatenulatum* Bi-OTA128 on DSS-induced colitis, which could be used as a candidate probiotic supporting recovery from ongoing colitis.

## 1. Introduction

Inflammatory bowel disease (IBD) is a long-term, chronic and relapsing disorder of the gastrointestinal tract characterized by intestinal inflammation and epithelial injury [1]. The pathogenic mechanisms of IBD are multifaceted but commonly associated with excessive oxidative stress, unbalanced gut microbiota, and aberrant immune responses [2]. Recently, there is a discernible rise in the prevalence of inflammatory bowel disease (IBD) on a global scale, with a noteworthy emergence of this trend in regions of Asia that have historically exhibited low rates of the disease [3]. Traditional therapies for mild-to-severe IBD include amino-salicylates, steroids, and TNF-α inhibitors, but with the limitations of side-effects and unsuitability for long-term treatment [4]. Biopharmaceutical agents and novel orally administered small molecules, such as anti-integrins, anti-cytokines, and Janus Kinase (JAK) inhibitors, can suppress the inflammatory activity and lead to the remission of IBD. Nevertheless, a comprehensive understanding of their efficacy, long-term safety profile, and cost-effectiveness has yet to be definitively elucidated [5]. Currently, probiotics as live biotherapeutic products are considered as novel, cost-effective therapeutic and preventive strategies for IBD treatment with minimal side effects [6,7].

The probiotics intervention can modulate gut microbial composition and improve the host–microbe interaction, thereby directly benefiting IBD patients [8,9]. In IBD patients, the abundance of bifidobacteria declines dramatically, while the supplementation of single or complex bifidobacterial preparation has been shown to moderate IBD symptoms and result in clinical or endoscopic remission [10,11,12]. Moreover, multiple *Bifidobacterium* strains have shown anti-inflammatory effects in a DSS-induced colitis mice model by enhancing the intestinal barrier, inhibiting the inflammatory responses, alleviating oxidative stress, and modulating the gut microbiota [13,14,15,16]. Notably, some exopolysaccharide (EPS)-producing *Bifidobacterium* strains with a higher EPS production ability exhibited better effects in colitis alleviation, indicating that EPS could act as a key effector in colitis treatment. For example, the *B. lactis* strain H4-2 with a higher EPS production showed a more effective repair function in DSS-induced colitis in mice than that of H9-3 [17]. Similarly, an EPS-producing recombinant *B. lactis* strain with a mucoid-ropy phenotype was found to regulate inflammatory responses in a DSS-induced colitis mouse model, but its non-ropy mutants showed no modulation effect [18]. The ropy EPS-producing strain *B. longum* YS108R also alleviated DSS-induced colitis by maintaining an intestinal barrier and modulating the gut microbiota, but not the non-ropy EPS-producing strain C11A10B [19]. It is noteworthy that the anti-inflammatory effects of EPS-producing strains are strain-specific due to the variations in EPS structure [20,21].

*B. pseudocatenulatum* Bi-OTA128 was a newly isolated EPS-producing strain from the fecal samples of a healthy elderly woman in Beijing, China, according to published methods [22]. The strain was stored in the China General Microbiological Culture Collection Center (CGMCC No. 28075). Previously, a study indicated that the Bi-OTA128 strain produced a high-molecular-weight EPS with a special *O*-acetyl group in its structure. The EPS component could inhibit the inflammatory responses by decreasing the NO, TNF-α, IL-1β, and IL-6 levels in lipopolysaccharide (LPS)-treated RAW264.7 cells (Appendix A). A non-EPS-producing strain *B. pseudocatenulatum* BLYR01-7 isolated from the same batch of fecal samples was chosen as a treatment comparison. In this study, we aimed to investigate the anti-inflammatory effects of two *B. pseudocatenulatum* strains on a DSS-induced colitis mice model. The beneficial effects of EPS-producing strain Bi-OTA128 were highlighted and its mechanisms were further clarified for possible in situ application in the gut.

## 2. Materials and Methods

### 2.1. Preparation of Bacterial Culture Suspension

Two bifidobacterial strains, *B. pseudocatenulatum* Bi-OTA128 and BLYR01-7, were sub-cultured three times by 3% inoculum in de Man, Rogosa, and Sharp medium supplemented with 0.05% (*w*/*v*) L-cysteine hydrochloride (MRSc) under anaerobic conditions. The bacterial pellet was collected by centrifugation at 4 °C and 8000× *g* for 5 min after cultivation at 37 °C for 24 h. Then, a fresh bacterial suspension was prepared by re-suspending the pellet to ~1.0 × 10^10^ CFU/mL with phosphate-buffered solution (PBS, pH 7.4) before administration.

### 2.2. Animals and Colitis Induced by DSS

Specific pathogen-free C57BL/6J mice (male, 6–8 weeks, 18–20 g, *n* = 32) were purchased from Beijing Huafukang Biotechnology Co., Ltd. (Beijing, China) and adapted for 7 days before experiment. All the mice were maintained in a barrier facility under standard laboratory conditions: 25 ± 2 °C, 50 ± 5% humidity, and 12 h light/dark cycle. After 1 week of acclimatization, the mice were randomly divided into four groups (*n* = 8 mice per group; Figure 1): the control group, the DSS model group (induced colitis without intervention), the DSS + Bi-OTA128 group (induced colitis with the pre-treatment of EPS-producing strain *B. pseudocatenulatum* Bi-OTA128), and the DSS + BLYR01-7 group (induced colitis with the pre-treatment of non-EPS-producing strain *B. pseudocatenulatum* BLYR01-7). Colitis was induced by adding 2.5% DSS (Mw: from 36,000 to 50,000 Da; MP Biomedicals) to drinking water, which was replaced every two days. From days 0 to 21, the mice in the DSS, DSS + Bi-OTA128, and DSS + BLYR01-7 groups were given normal saline (0.2 mL per mouse) or bacterial suspension (2 × 10^9^ CFU in 0.2 mL saline per mouse) by oral gavage on a daily basis. During this period, colitis was induced by drinking 2.5% DSS solution during days 14-21. The control group was given the same amount of normal saline by drinking sterilized water. All of the mice were sacrificed by cervical dislocation at the end of day 21. The study was approved by the Institutional Animal Care and Use Committee of the China Agricultural University, Beijing, China (No. AW03503202-1), and all of the animal handling and experimental procedures were performed in accordance with the National Research Council’s Guide for the Care and Use of Laboratory Animals.

### 2.3. Colitis Assessment and Histopathology Analysis

The percentage of body weight (BW) loss of each mouse was calculated relative to the initial weight prior to DSS treatment. The disease activity index (DAI) score was assessed daily during DSS treatment according to the body weight loss, stool consistency, and stool occult blood (Appendix A). After sacrifice, the colonic length and the percentage of spleen in body weight were also measured in this study.

The distal colon of each mouse, 1 cm from the anus, was taken and fixed in Carnoy’s solution for histopathological analysis. After being embedded in paraffin wax, the colon samples were cut into 5 μm sections and stained with hematoxylin and eosin (H&E). Six randomly selected fields of each sample were assessed according to the inflammation severity, inflammation extent, and crypt damage based on our previous study [16]. The detailed scoring criteria for colonic histopathology was summarized in Appendix A. Moreover, the Alcian blue (AB) staining to observe mucus distribution in the colon and the periodic acid–Schiff (PAS) staining to count goblet cells were also performed as described previously [23].

### 2.4. Determination of Biochemical Indices in Colon

The colon tissues (~50 mg per mouse) were homogenized in precooled RIPA lysis buffer (Beyotime Biotechnology, Shanghai, China) at a ratio of 1:10 (*w*/*v*) with sterilized glass beads using a SCIENTZ-48L high-throughput tissue grinder (Ningbo Scientz Biotechnology, Zhejiang, China). After centrifugation at 4 °C, 12,000× *g* for 10 min, the supernatants were collected and the protein concentration in each sample was determined by BCA Protein Assay Kit (Beyotime Biotechnology). The oxidative stress indices of malondialdehyde (MDA), superoxide dismutase (SOD), catalase (CAT), and glutathione peroxidase (GSH-Px) in colon tissues were determined by commercial biochemical assay kits (Nanjing Construction Co., Ltd., Nanjing, China) according to the manufacturer’s instructions. The inflammatory cytokines of TNF-α, IL-1β, IL-6, and IL-10 in colon tissues were measured by ELISA kits (Elabscience Biotechnology Co. Ltd., Wuhan, China).

### 2.5. Quantitative Real-Time PCR Analysis

The colon tissues (~20 mg per mouse) were washed with cold PBS (pH 7.4) and stored in RNAlater (Sigma-Aldrich, Shanghai, China) at −20 °C for RNA isolation. Total RNA from each colonic tissue sample was extracted by the FastPure^®^ Cell/Tissue Total RNA Isolation Kit (Vazyme Biotech Co., Ltd., Nanjing, China) according to the manufacturer’s instructions. The RNA quantity and quality were then accurately assessed by the Qubit RNA BR Assay Kit (Invitrogen, Eugene, OR, USA) and NanoDrop ONE^c^ Spectrophotometer (Thermo Fisher Scientific, Waltham, MA, USA). Subsequently, 1 μg of total RNA was used to generated the complementary DNA by HiScript III All-in-one RT SuperMix Perfect for qPCR Kit with gDNA remover (Vazyme). Quantitative real-time polymerase chain reaction (RT-qPCR) was then performed using the High ROX Premixed ChamQ SYBR Color qPCR Master Mix (Vazyme) with Step One Plus Real-time PCR System (Applied Biosystems, CA, USA). All the primers used in this study are listed in Table 1. The relative quantification of the target gene was calculated by comparing with the β-actin as reference according to the 2^−ΔΔCT^ method [24].

### 2.6. Gut Microbiota Analysis

Fecal samples were collected by placing mice separately in vacuum stem-sterilized cages to defecate naturally and the feces were stored at −80 °C. The total microbial DNA in feces were extracted by using the Magnetic Stool DNA Kit (Tiangen Biotech Co., Ltd., Beijing, China). The V3–V4 regions of the bacterial 16S rRNA gene were amplified with the primers 341F (5′-CCTAYGGGRBGCASCAG-3′) and 806R (5′-GGACTACNNGGGTATCTAAT-3′). After being purified with Universal DNA Purification Kit (Tiangen Biotech) for PCR products, the sequencing library was generated using NEBNext^®®^ Ultra^TM^ II FS DNA PCR-free Library Prep Kit (New England Biolabs, Ipswich, MA, USA) in accordance with the manufacturer’s instructions. Then, paired-end reads sequencing (2 × 250) was performed on an Illumina Novaseq 6000 platform at Novogene Bio-informatics Co., Ltd., Beijing, China.

Raw sequencing data were further processed using the QIIME2 software (version QIIME2-202202). Briefly, amplicon sequence variants (ASVs) were processed by the DADA2 tool (version 0.99.8) [25] and then the species were annotated using the Silva database [26]. The Alpha diversity of the microbiota was evaluated with Chao1, Shannon, and Simpson index. Beta diversity was assessed by hierarchical clustering tree, principal coordinate analysis (PCoA), and non-metric multidimensional scaling analysis (NMDS) based on the weighted unifrac distance [27]. For community difference analysis, MetaStat based on multiple hypothesis test was performed in R with Complex Heatmap package (version 2.19.0). The linear discriminant analysis effect size (LEfSe) analysis was also performed to assess the taxonomic differences between the groups [28]. The differentially abundant taxa were identified by the Tukey and Kruskal–Wallis H test with *p* < 0.05.

### 2.7. Statistical Analysis

The results in this study were expressed as the mean ± standard deviation (SD). The statistical differences among four experimental groups were determined by one-way analysis of variance (ANOVA), followed by Tukey’s post hoc test using GraphPad Prism software (version 8.4.3). The mean values of the treatment group versus the control group were compared by Dunnett’s *t*-test. Values of *p* < 0.05 were considered to be statistically significant.

## 3. Results

### 3.1. B. pseudocatenulatum Intervention Improved Colitis Symptoms

The experimental design to assess the alleviation effects of *B. pseudocatenulatum* strains on DSS-induced colitis in C57BL/6J mice was illustrated in Figure 1. Generally, the body weight loss is a direct manifestation of DSS-induced colitis in mice. In this study, the DSS treatment led to nearly 18% body weight loss compared with the Control group. Pre-treatment with *B. pseudocatenulatum* Bi-OTA128 and BLYR01-7 decreased the body weight loss to 5.9% and 11.7%, respectively (Figure 2A). The DAI score of the DSS group reached 9.75 (total score of 11), indicating severe colitis symptoms in DSS-treated mice. However, this value decreased by 41.44% and 14.05% after the treatment by Bi-OTA128 and BLYR01-7 strains, respectively, which suggested that the *B. pseudocatenulatum* supplement improved the DSS-induced colitis in mice (Figure 2B).

The DSS group mice showed the shortest colonic length (5.39 cm) and the largest percentage of spleen in body weight (0.54%, Figure 2C–E). The colonic lengths of the Bi-OTA128 and BLYR01-7-treated mice were 1.24 and 1.06 times longer than that of DSS-treated mice (*p* < 0.05; Figure 3D). The ratio of the spleen/body weight in Bi-OTA128 treated mice was also significantly decreased to 0.43%; however, the BLYR01-7-treated mice showed no difference (Figure 2E). Taken together, the *B. pseudocatenulatum* intervention improved the symptoms of DSS-induced severe colitis, and the EPS-producing strain Bi-OTA128 showed better effects than that of the non-EPS-producing strain BLYR01-7, indicating that the production of EPS may give advantages in its probiotic property.

### 3.2. B. pseudocatenulatum Alleviated Colon Damage and Recovered Mucous Layer

The colon of control group mice showed tightly arranged epithelial cells, intact intestinal fold and crypt structure, and no inflammatory cell infiltration. In contrast, the DSS treatment caused mucosal and submucosal edema, crypt loss, and severe inflammatory cell infiltration (Figure 3A, row 1). The histological score of the DSS group mice also reached the highest value of 9.0 (total score of 10). Pre-treatment with *B. pseudocatenulatum* strains Bi-OTA128 and BLYR01-7 alleviated the colon damage and reduced 46.33% and 20.34% of histological score than that of DSS-treated mice, respectively (*p* < 0.01; Figure 3B). Compared with the BLYR01-7 strain intervention, mice with Bi-OTA128 treatment showed mild crypt loss and cell inflammation, and exhibited a better histological performance.

The PAS staining (PAS^+^) goblet cells were evenly distributed in the crypt structure of control group mice (Figure 3A, row 2), and the number reached 16.7 PAS^+^ cells per crypt (Figure 3C). Due to severe crypt loss and cell infiltration, the PAS^+^ cells in the DSS group mice were only 4.2 per crypt, which was 74.85% lower than that of the control mice (*p* < 0.0001; Figure 3C). Meanwhile, the DSS treatment also severely destroyed the mucous layer (Figure 3A, row 3) and simultaneously reduced the MUC2 gene expression (Figure 3D). Pre-treatment with *B. pseudocatenulatum* strains Bi-OTA128 and BLYR01-7 significantly increased the PAS^+^ cells by 2.92 and 1.44 times compared with the DSS group mice (Figure 3C), and the MUC2 gene were 4.36-fold and 3.61-fold up-regulated, respectively (Figure 3D). These results indicated that the *B. pseudocatenulatum* intervention could alleviate the DSS-induced colon damage and recover the mucous layer, whilst the EPS^+^ strain Bi-OTA128 showed better effects than that of EPS^−^ strain BLYR01-7.

### 3.3. B. pseudocatenulatum Attenuated Oxidative Stress in Colon Tissues

After treatment with DSS, the MDA level was dramatically increased to 7.16 nmol/mg protein in the DSS mouse group, which was 1.71-fold higher than that in control group (Figure 4A). While the SOD, CAT, and GSH-Px activities were significantly decreased by 36.92%, 45.92%, and 53.33% in DSS group mice compared with the control (Figure 4B–D). Notably, the Bi-OTA128 and BLYR01-7 intervention significantly lowered the MDA levels, but increased the SOD, CAT, and GSH-Px activities in colon tissues relative to the DSS group (*p* < 0.05). Although the difference was not significant in the SOD activity between the Bi-OTA128 and BLYR01-7 intervention, the lower MDA level and higher CAT and GSH-Px activities in the colon tissues indicated a better attenuation effect on the oxidative stress of the EPS^+^ strain Bi-OTA128.

### 3.4. B. pseudocatenulatum Improved Gut Barrier Integrity and Enhanced Tight Junctions Expression

The mRNA levels of three tight junction proteins ZO-1, occludin, and claudin-1 were down-regulated in DSS-treated mice, which were 0.55-, 0.46-, and 0.36-fold compared with the control mice, respectively (Figure 5A–C). This result suggested that the DSS treatment significantly damaged gut barrier integrity, which was consistent with what has been observed in H&E staining (Figure 3A). However, the *B. pseudocatenulatum* Bi-OTA128 and BLYR01-7 intervention up-regulated the gene expression of ZO-1, occludin, and claudin-1 by 1.55~2.54-, 1.70~2.04-, and 1.68~2.67-fold compared with the DSS mouse group, which suggested that pre-treatment with *B. pseudocatenulatum* helps to improve DSS-induced gut barrier damage (Figure 5A–C). Moreover, two Toll-like receptors, namely TLR2 and TLR4, were found to be highly expressed in DSS-treated mice, leading to the overactive identification of intestinal environmental factors. Intervention with the Bi-OTA128 strain significantly decreased the TLR2 and TLR4 mRNA expressions and the level of TLR 4 was similar to that in control group mice (Figure 5D,E). Taken together, the *B. pseudocatenulatum* treatment could improve gut barrier integrity by enhancing the expression of tight junctions and simultaneously suppress the over-expression of the relative recognition receptors of TLR2/4.

### 3.5. B. pseudocatenulatum Regulated Inflammatory Response in Colon Tissues

In this study, DSS treatment triggered dramatically increased levels of pro-inflammatory cytokines, such as TNF-α, IL-1β, and IL-6 in colon tissues, which were 4.49, 3.93, and 3.55 times higher than those in control mice, respectively (*p* < 0.0001; Figure 6A–C). However, the anti-inflammatory cytokine IL-10 in the DSS mouse group was decreased 45.85% compared with the control mice (*p* < 0.0001; Figure 6D), indicating that an excessive immune response occurred with DSS treatment. Notably, the TNF-α, IL-1β, and IL-6 levels in Bi-OTA128-treated mice were 67.16%, 63.11%, and 49.87% lower than those of the DSS mouse group (Figure 6A–C), but the IL-10 levels were significantly increased by 1.36 times, which suggested that the Bi-OTA128 intervention could regulate the immune response to a balanced state. Moreover, the supplementation of Bi-OTA128 showed advantages in the regulation of inflammatory responses compared to the BLYR01-7 strain with regard to the lower pro-inflammatory cytokine levels of TNF-α, IL-1β, and IL-6 in colon tissues, although the difference in IL-10 levels was not significant (*p* > 0.05).

### 3.6. B. pseudocatenulatum Modulated Gut Microbiota

The Veen diagram showed 217 shared ASVs among the four experimental groups, but 417, 134, 180, and 296 specific ASVs in the control (C1), DSS (D1), DSS + Bi-OTA128 (B1), and DSS + BLYR01-7 (B2) groups, respectively, which suggested differential gut microbiota composition in the tested groups (Appendix A). The alpha-diversity (Chao1, Shannon, and Simpson index) of gut microbiota showed no differential within-community between the control and DSS group (Figure 7A–C). However, the EPS-producing strain Bi-OTA128 intervention significantly increased the community diversity of the gut microbiota compared with the DSS group (*p* < 0.05; Figure 7B). Meanwhile, the Beta-diversity, assessed by NMDS and PCoA, was shown to be a distinctly clustered community among the four experimental groups (stress value = 0.124; Figure 7D). Notably, the distribution of gut microbiota in the Bi-OTA128 treatment group was much closer to the control group compared with BLYR01-7-treated mice (Appendix A). These results indicated that the DSS treatment or *B. pseudocatenulatum* intervention changed the gut microbiota composition in the different forms in DSS-induced colitis mice model. Then, the differential abundance taxa in individual groups were further clarified.

MetaStat analysis showed that DSS treatment significantly reduced the relative abundance of *Bacteroidota* and *Actinobacteriota* in the Phylum level; however, the *Proteobacteria* and *Firmicutes* were highly increased compared with the control mice (*p* < 0.05). The *B. pseudocatenulatum* intervention increased the relative abundance of *Actinobacteriota* compared with the DSS group, and the Bi-OTA128 treatment also increased the *Bacteroidota* but decreased the *Proteobacteria* abundance compared with the DSS-treated mice (Figure 7E). At the family level, the relative abundance of *Muribaculaceae* and *Bifidobacteriaceae* in the DSS mouse group was significantly reduced, but the *Enterobacteriaceae* and *Erysipelotrichaceae* were conversely increased compared with the control mice (D1 vs. C1; Appendix A). Notably, the Bi-OTA128 treatment clearly changed the gut microbiota composition by reducing the increased abundance of *Enterobacteriaceae* and *Erysipelotrichaceae* in the DSS group mice, and simultaneously increasing the *Muribaculaceae* and *Bifidobacteriaceae* abundance, which were highly decreased in the DSS mouse group (B1 vs. D1; Appendix A).

Combined with the histogram and cladogram of LEfSe analysis (LDA score > 4.0; Appendix A), the differentially abundant taxa in the Genus level are summarized in Figure 7F. Compared with the control mice, the relative abundance of twelve genera in DSS-treated mice were significantly increased, including *Escherichia-Shigella*, *Enterorhabdus*, *Halomonas*, *Romboutsia*, *Turicibacter*, *Candidatus_Saccharimonas*, *Parvibacter*, *Lactococcus*, *Enterobacter*, *Lachnospiraceae_NK4A126*, and *Oscillibacter*. However, three genera *Bifidobacterium*, *Lactobacillus*, and *Lachnospiraceae_A2* were highly decreased (*p* < 0.05). Importantly, the EPS-producing strain Bi-OTA128 modulated gut microbiota involving multiple genera and showed a differential influence compared to BLYR01-7, a non-EPS-producing strain. Although the Bi-OTA128 and BLYR01-7 treatments both increased the *Bifidobacterium* and decreased the *Escherichia-Shigella* abundance compared with DSS mice, the Bi-OTA128 intervention also down-regulated *Enterorhabdus*, *Halomonas*, *Ligilactobacillus*, *Romboutsia*, but up-regulated *Candidatus_Saccharimonas* and *Monoglobus*, which were not differentially abundant in the B0-17 group mice. Moreover, the relative abundance of the three genera *Enterococcus*, *Lactococcus*, and *Enterobacter* were significantly decreased after the Bi-OTA128 treatment, but were highly increased in the BLYR01-7 mouse group, which indicated the strain diversity of the probiotic intervention in the modulation of gut microbiota. Taken together, the *B. pseudocatenulatum* intervention could the modulate DSS-induced gut microbiota dysbiosis, and the EPS-producing strain Bi-OTA128 treatment showed a similar gut microbiota composition of the control mice.

## 4. Discussion

In the present study, we found that the supplementation of the EPS-producing strain Bi-OTA128 showed better effects on the alleviation of DSS-induced severe colitis symptoms than that of the non-EPS-producing strain BLYR01-7 (Figure 2). Meanwhile, the alleviation effects of the Bi-OTA128 strain were comparable with a reported conjugated linoleic acid (CLA)-producing strain *B. pseudocatenulatum* MY40C but better than that of CCFM680 strain [14] based on body weight loss, DAI score, and colon length, indicating that Bi-OTA128 could be a candidate probiotic for colitis treatment. The mechanisms of EPS-producing bifidobacterial strains involved in IBDs are multifaceted, mainly including the improvement of the intestinal barrier integrity, balancing the intestinal microbial composition, and regulating immune-related cytokine expression [29].

The mucus layer and epithelial cells construct the first physical barrier to defense the invasion of luminal contents, but are heavily damaged in IBD patients [30,31]. In this study, the DSS treatment significantly inhibited the MUC2 expression, a predominant component of mucus in the colon, and reduced the number of goblet cells in crypt, which resulted in a damaged mucus layer in the DSS group mice. The *B. pseudocatenulatum* intervention reversed the DSS-induced mucus layer damage and showed an intuitive improvement observed by PAS and AB staining (Figure 3A). Moreover, the expression of ZO-1, occludin, and claudin-1 were also highly upregulated at transcriptional level after *B. pseudocatenulatum* treatment compared with the DSS group (Figure 5A–C), indicating that the bifidobacterial intervention helped enhance the integrity of the epithelial monolayer, a critical indicator for an intestinal mechanical barrier [32]. These results were consistent with what has been reported in *B. breve* H4-2 and H9-3, *B. lactis* A6 and *B. pseudocatenulatum* MY40C supplementation in a DSS-induced colitis [14,16,17].

An imbalance in redox homeostasis is a major inducing factor in the initiation and progression of IBD, which triggers oxidative stress in the gastrointestinal tract by excessive reactive oxygen species (ROS) generation [33]. Oxidative stress can accelerate epithelial cell damage by modifying the protein functions and causing lipid peroxidative, resulting in the destruction of the mucosal layer in the GI tract and bacterial invasion, which in turn stimulate the immune responses [34]. In this study, the DSS treatment significantly increased the MDA contents in the colon tissues, a biomarker of lipid peroxidation in damaged tissues. However, the *B. pseudocatenulatum* treatment reduced the colonic MDA levels and showed a protective effect against cell peroxidation (Figure 4A). Moreover, the three intracellular enzymatic antioxidants of SODs, CAT, and GSH-Px in the colon tissues of DSS-treated mice were highly decreased compared with the control mice (Figure 4B–D). These constitute the endogenous antioxidant defense system in a body to counteract the uncontrolled oxidative stress caused by excess ROS [33]. The supplementation of *B. pseudocatenulatum* significantly increased the SODs, CAT, and GSH-Px contents in the colon tissues compared with the DSS-treated mice, indicating that the bifidobacterial intervention could enhance the colonic antioxidant ability. The beneficial effects are comparable with a bifidobacterial strain *B. lactis* A6 on DSS-induced colitis according to our previous study [16], which revealed a possible mechanism of alleviating DSS-induced colitis from an antioxidant perspective.

An unbalanced intestinal immune system is another characteristic of IBD [35]. The epithelial cell damage-induced bacterial invasion and excessive ROS-caused oxidative stress could aggravate the progression of inflammation responses in colon tissues, and promote the over-production of pro-inflammatory cytokines such as TNF-α, IL-1β, and IL-6, by dendritic cell, macrophage, neutrophil or effector T cells [36]. The three pro-inflammatory cytokines were dramatically increased in DSS-treated mice, but were highly decreased after *B. pseudocatenulatum* treatment (Figure 6A–C), especially in the Bi-OTA128-treated mice. This phenomenon indicated that the bifidobacterial intervention could improve the immune response disorder in DSS-induced colitis and the EPS-producing strain showed better effects. Similar results were reported that the administration of *B. breve* H4-2 or H9-3 decreased TNF-α, IL-1β, and IL-6 levels in a DSS-induced mouse model; however, the strain H4-2 with a higher ability to produce EPS was more effective than H9-3 [17]. The production of EPS may give its bacterium advantages in persistence in the gut and facilitate constant actions on the host–microbe interactions [20]. Moreover, the *B. pseudocatenulatum* treatment increased the levels of the anti-inflammatory cytokine IL-10 in colon tissues (Figure 6D), indicating that bifidobacterial intervention could regulate the immune response to a balanced situation by attenuating mucosal inflammation.

In particular, two pattern recognition receptors, TLR2 and TLR4, were also found to be highly down-regulated after *B. pseudocatenulatum* treatment compared with the DSS group (Figure 5D,E). TLR2 and TLR4 participate in the recognition of microbial components in the gut and the activation of the downstream NF-κB inflammatory pathway [37]. The down-regulated TLRs also portend a changed gut microbiota composition after bifidobacterial treatment. In this study, the DSS treatment increased the relative abundance of bacteria involved in twelve genera, including the maleficent bacteria of *Escherichia-Shigella*, *Enterorhabuds*, *Turicibacter*, *Enterobacter*, and *Osillibacter*, which were associated with the induction of the intestinal inflammation [17,19]. However, the beneficial bacteria of *Bifidobacterium* and *Lactobacillus* were highly decreased in DSS-treated mice, which showed protective effects against colonic inflammation with the production of short-chain fatty acids (SCFAs) in the gut [38]. *B. pseudocatenulatum* intervention significantly increased the relative abundance of *Bifidobacterium* but reduced *Escherichia-Shigella* in the gut, indicating a modulation effect in gut dysbiosis. Notably, the EPS-producing strain Bi-OTA128 also reduced the relative abundance of *Enterorhabuds*, *Halomonas*, *Romboutsia*, *Lactococcus*, and *Enterobacter,* which highly increased in the DSS-treated mice and recovered the gut microbiota composition to a balanced situation similar to the control mice (Figure 7F). Interestingly, the genera *Clostridium sensu stricto* was only enriched in Bi-OTA128 treated mice, which benefits the development of the immune system by degrading polysaccharides to generate acetic acid and butyrate [39,40]. This result suggested that the supplementation of the EPS-producing strain Bi-OTA128 induced cross-feeding between microbiota, which provides a new insight into the alleviation of intestinal inflammation.

Collectively, this study reported the advantages of the beneficial effects of an EPS-producing strain *B. pseudocatenulatum* Bi-OTA128 on DSS-induced colitis than a non-EPS-producing strain BLYR01-7. Multiple mechanisms were involved in the alleviation effects of the EPS-producing strain *B. pseudocatenulatum* Bi-OTA128 on DSS-induced colitis. As illustrated in Figure 8, the Bi-OTA128 treatment enhanced the physical barrier, reduced the oxidative stress, regulated the immune responses, and modulated the gut microbiota to resist DSS-induced colonic damage. These results indicated that the produced EPS may act as a key functional component to support the recovery from ongoing colitis. Regrettably, the EPS deficiency Bi-OTA128 mutant strain was failed to construct. In order to further confirm the function of EPS, the method of the genetic transformation needs to be solved in the future.

## 5. Conclusions

In this study, the *B. pseudocatenulatum* intervention alleviated the DSS-induced colitis in mice, and the EPS-producing strain Bi-OTA128 showed the better effects than that of non-EPS-producing strain BLYR01-7. Compared with the DSS group mice, the Bi-OTA128 treatment significantly alleviated the DSS-induced colonic damage and increased the MUC2 and TJs expressions to enhance intestinal barrier. Antioxidant enzymes including SOD, CAT, and GSH-Px, were also enhanced to reduce the DSS-induced oxidative stress. Moreover, the pro-inflammatory cytokines of TNF-α, IL-1β, and IL-6 were decreased, and the anti-inflammatory cytokine IL-10 was increased after treatment, leading to a balanced immune homeostasis. Furthermore, the gut microbiota analysis demonstrated that the Bi-OTA128 treatment specifically increased the relative abundance of genera *Clostridium sensu stricto*, which indicated a cross-feeding effect in modulating gut microbiota. This study provides important experimental evidence for the role of an EPS-producing *B. pseudocatenulatum* strain on the alleviation of DSS-induced colitis. Taken all together, the Bi-OTA128 strain could be used as a candidate probiotic supporting recovery from ongoing colitis.

## Figures and Tables

**Figure 1 nutrients-15-04993-f001:**
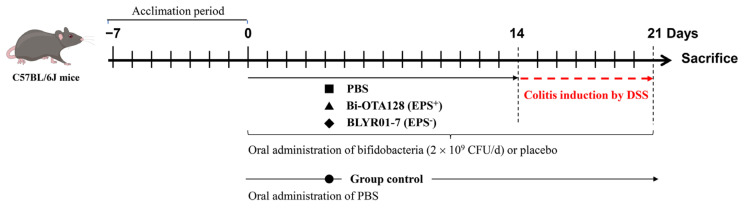
Experimental design to assess the alleviation effects of *B. pseudocatenulatum* strains on dextran sodium sulfate (DSS)-induced colitis in C57BL/6J mice. PBS, phosphate-buffered solution; EPS^+^, EPS-producing bacteria; EPS^−^, non-EPS-producing bacteria.

**Figure 2 nutrients-15-04993-f002:**
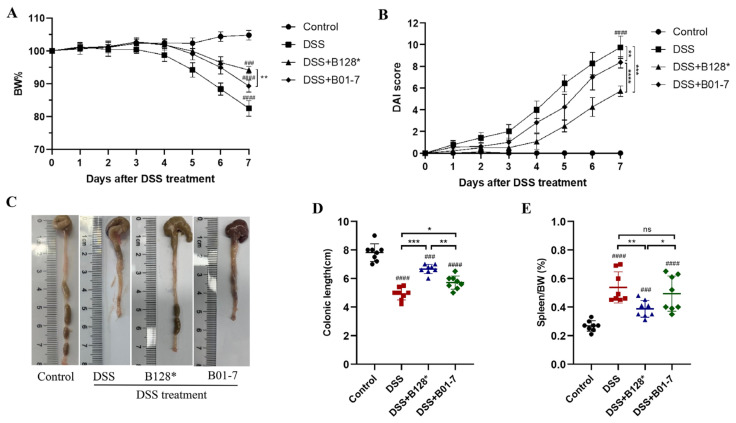
Effects of the *B. pseudocatenulatum* treatment on the symptoms of DSS-induced colitis in C57BL/6J mice: (**A**) body weight change; (**B**) DAI score; (**C**) representative colon images; (**D**) colonic length; and (**E**) the ratio of the spleen in the body weight of mice. *n* = 8 mice per group. ### *p* < 0.001 and #### *p* < 0.0001 showed a significant difference compared with the control group by Dunnett’s test. * *p* < 0.05, ** *p* < 0.01, *** *p* < 0.001, and **** *p* < 0.0001 showed a significant difference with ANOVA followed by Tukey’s post hoc test. ns, non-significance; B128*, EPS-producing strain Bi-OTA128; B01-7, non-EPS-producing strain BLYR01-7.

**Figure 3 nutrients-15-04993-f003:**
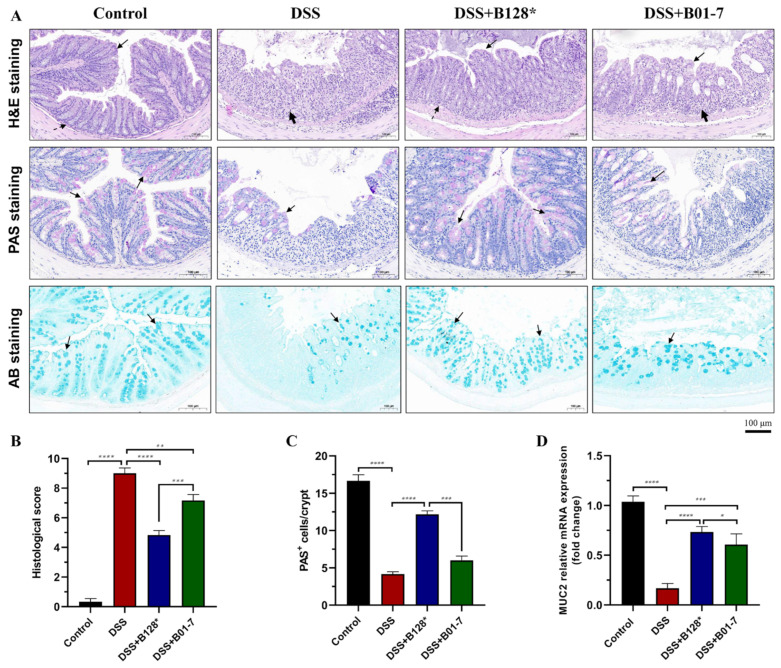
Effects of *B. pseudocatenulatum* intervention on histological injury, goblet cells distribution, and mucus secretion. (**A**) Representative hematoxylin and eosin (H&E), Alcian blue (AB), and the periodic acid–Schiff (PAS) staining of colon sections in each group (scale bar, 100 μm). In H&E staining, the black arrow, dashed arrow, and bold arrow point to the epithelial cell, crypt, and inflammatory cell infiltration, respectively. In PAS staining, the goblet cells in crypt were stained with red and pointed by black arrow. In AB staining, the mucus layer was stained with blue and marked with a black arrow. (**B**) Histological score, (**C**) PAS^+^ cells counting per crypt. Data were collected by randomly selecting six fields of each sample. (**D**) The relative mRNA expression of the MUC2 gene in colonic tissues normalized by β-actin. *n* = 8 mice per group. * *p* < 0.05, ** *p* < 0.01, *** *p* < 0.001, and **** *p* < 0.0001. B128*, EPS-producing strain Bi-OTA128; B01-7, non-EPS-producing strain BLYR01-7.

**Figure 4 nutrients-15-04993-f004:**
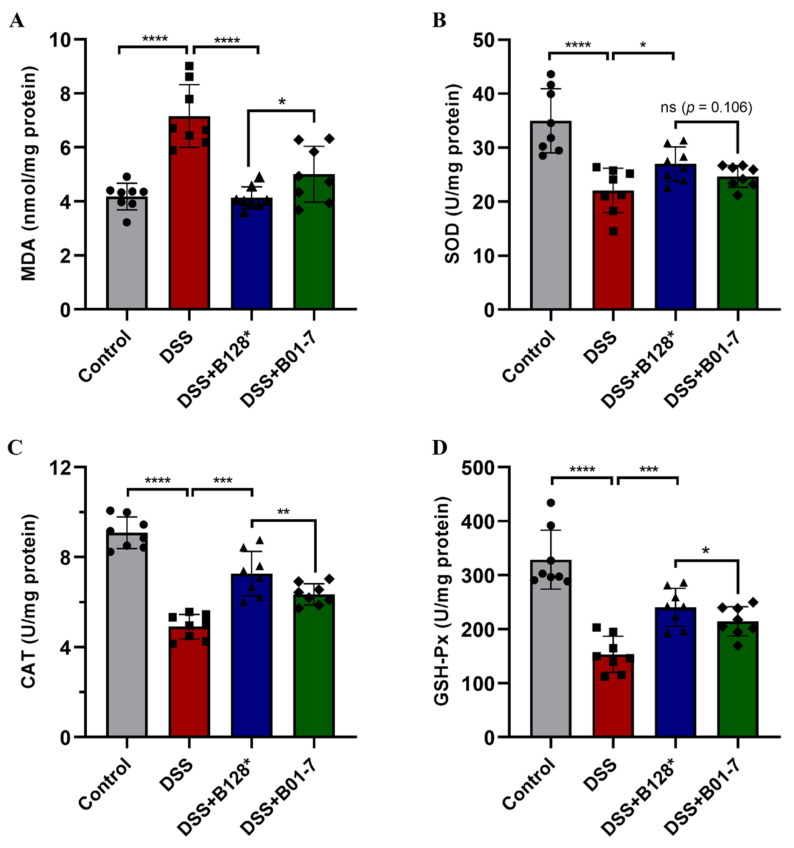
Effects of *B. pseudocatenulatum* treatment on oxidative stress in DSS-induced colitis. (**A**) MDA level; (**B**) SOD activity; (**C**) CAT activity; and (**D**) GSH-Px activity in colon tissues. *n* = 8 mice per group. Symbols of circle, square, triangle, and diamond show determined results from single mouse belonging to the Control, DSS, DSS+B128*, and DSS+B01-7 groups, respectively. ns, non-significance; * *p* < 0.05, ** *p* < 0.01, *** *p* < 0.001, and **** *p* < 0.0001. B128*, EPS-producing strain Bi-OTA128; B01-7, non-EPS-producing strain BLYR01-7.

**Figure 5 nutrients-15-04993-f005:**
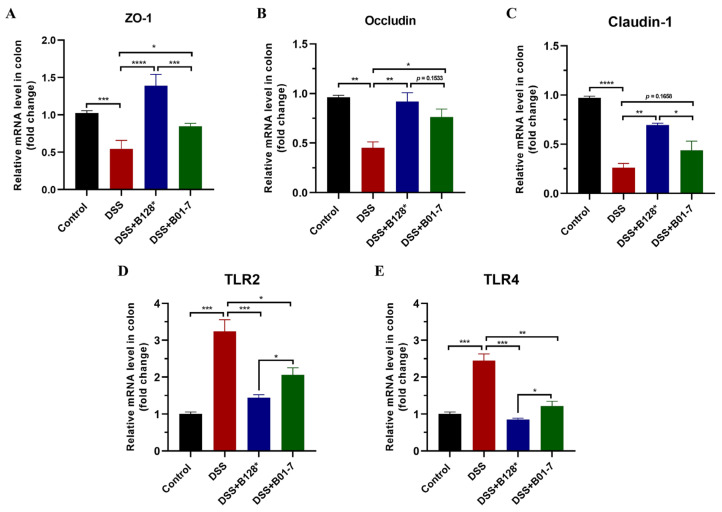
Effects of *B. pseudocatenulatum* treatment on gut barrier in DSS-induced colitis. The mRNA levels of (**A**) ZO-1, (**B**) Occludin, (**C**) Claudin-1, (**D**) TLR2, and (**E**) TLR4 in colon tissues. *n* = 3. * *p* < 0.05, ** *p* < 0.01, *** *p* < 0.001, and **** *p* < 0.0001. ZO-1, zonula occludens-1; TLR2, Toll-like receptor 2; TLR4, Toll-like receptor 4; B128*, EPS-producing strain Bi-OTA128; and B01-7, non-EPS-producing strain BLYR01-7.

**Figure 6 nutrients-15-04993-f006:**
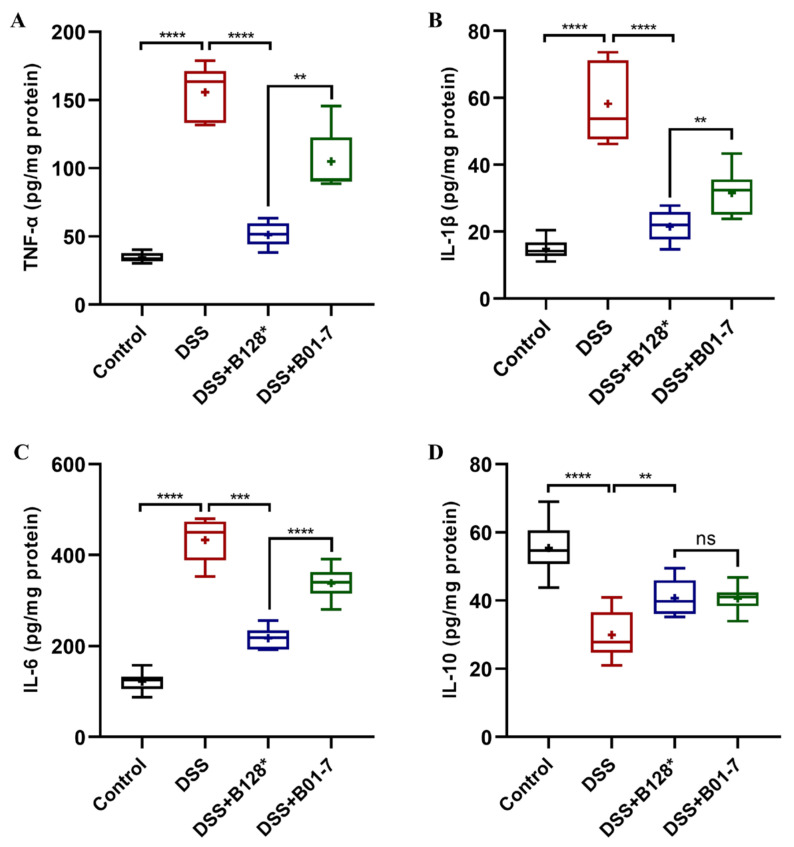
Effects of *B. pseudocatenulatum* intervention on inflammatory responses in mice with DSS-induced colitis: (**A**) TNF-α, (**B**) IL-1β, (**C**) IL-6, and (**D**) IL-10 levels in colon tissues. *n* = 8 mice per group. ns, no significant difference, ns, non-significance; ** *p* < 0.01, *** *p* < 0.001, and **** *p* < 0.0001. B128*, EPS-producing strain Bi-OTA128; and B01-7, non-EPS-producing strain BLYR01-7.

**Figure 7 nutrients-15-04993-f007:**
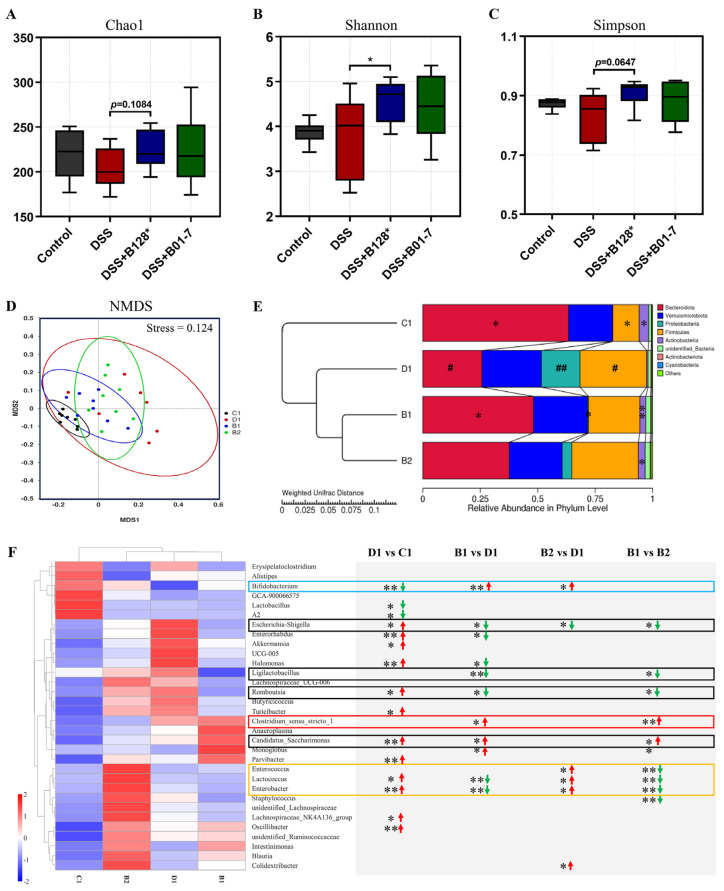
Effects of *B. pseudocatenulatum* intervention on gut microbiota composition in DSS-induced colitis in mice. (**A**–**C**) Chao1, Shannon, and the Simpson index of Alpha diversity. (**D**,**E**) NMDS and hierarchical clustering tree based on the weighted unifrac distance. The relative abundance of microbiota was displayed in the phylum level. # *p* < 0.05 and ## *p* < 0.01 represent comparison with the C1 (Control) group; * *p* < 0.05 and ** *p* < 0.01 represent the comparison with the D1 (DSS) group. (**F**) Heatmap of top 35 genera. The statistically significant difference was analyzed by the Tukey and Kruskal–Wallis H test. * *p* < 0.05, and ** *p* < 0.01 represent differentially abundant taxa between two groups. Green and red arrows show decreased and increased taxa with significance, respectively. NMDS, non-metric multidimensional scaling analysis; C1, the control group; D1, the DSS treatment group; B1, the DSS + Bi-OTA128 group; B2, the DSS + BLY01-7 group.

**Figure 8 nutrients-15-04993-f008:**
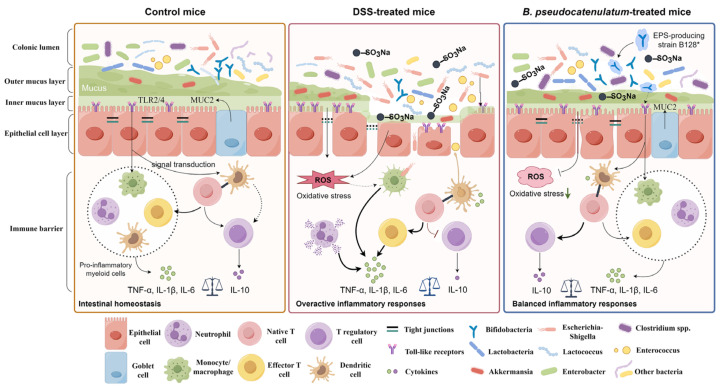
Proposed schematic of the alleviation effects of the EPS-producing strain *B. pseudocatenulatum* Bi-OTA128 in DSS-induced colitis mice model. The figure was drawn by Figdraw (version 2.0).

**Table 1 nutrients-15-04993-t001:** Primers used for RT-qPCR.

Target Protein	Function	Primer (5′→3′)	Product Size
ZO-1	Gut barrier integrity	F: GGCCTTGGCCTAGCATACAC	158 bp
		R: GTCTTCATTTGACCCTCCCTC	
Occludin		F: TCACTTTTCCTGCGGTGACTT	136 bp
		R: GGGAACGTGGCCGATATAAT	
Claudin-1		F: AGCTGTGCATGGCCTCTTGT	128 bp
		R: CCAATGTCAATGGCAACACCC	
MUC2	Mucus layer	F: TGCTGACGAGTGGTTGGTGAAT	135 bp
		R: GATGAGGTGGCAGACAGGAGAC	
TLR2	Pattern recognition receptor	F: GACTCTTCACTTAAGCGAGTCT	102 bp
		R: AACCTGGCCAAGTTAGTATCTC	
TLR4		F: GCCATCATTATGAGTGCCAATT	107 bp
		R: AGGGATAAGAACGCTGAGAATT	
β-actin	Reference	F: CCTAAGAGGAGGATGGTCGC	230 bp
		R: CTCAACACCTCAACCCCCTC	

## Data Availability

The data presented in this study are available upon request from the author.

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
