# Peer review of "A Ropy Exopolysaccharide-Producing Strain Bifidobacterium pseudocatenulatum Bi-OTA128 Alleviates Dextran Sulfate Sodium-Induced Colitis in Mice"

_nutrients, 2023, doi:10.3390/nu15234993_

Round 1

Reviewer 1 Report

Comments and Suggestions for Authors

In this study, Hui Wang, et al. investigated the anti-inflammation effects of an exopolysaccharide (EPS)-producing strain Bifidobacterium pseudocatenulatum Bi-OTA128 through a dextran sulfate sodium (DSS)-induced colitis mice model. They conclude that B. pseudocatenulatum treatment improved DSS-induced colitis symptoms and maintained intestinal barrier integrity by up-regulating MUC2 and tight junctions’ expression.

Strengths of the study:

-        Study question is valid

-        Adequate literature review was performed. References are appropriate.

-       The research results validate the author's conclusions.

The manuscript can be improved by addressing the following concerns.

- Studies have shown that EPS producing strains of Bifidobacterium improved DSS-induced colitis (Eg; A ropy exopolysaccharide producing strain Bifidobacterium longum subsp. longum YS108R alleviates DSS-induced colitis by maintenance of the mucosal barrier and gut microbiota modulation. Food Funct., 2019,10, 1595-1608). Authors may want to discuss clearly regarding the beneficial effects of their study strain as compared to similar EPS producing strains of Bifidobacterium.

- Authors may want to explain what is unique in their study methods as compared to previous publications. It appears that authors used same study methods with different probiotic strains. Consider analyzing probiotic characteristics of strains, use of positive control, and measure short chain fatty acid levels.

- Include limitations of their study.

- There have been minor English language errors in several places. Consider careful revision with attention to grammar to improve readability.

Examples: Previously study showed that the Bi-OTA128 strain produced a high-molecular-weight EPS with a special O-acetyl group in its structure, which could inhibit inflammatory responses by decreasing NO, TNF-α, IL-1β, and IL-6 levels in lipopolysaccharide (LPS)-treated RAW264.7 cells (Figure S1). Line 82 onwards…

Comments on the Quality of English Language

There have been minor English language errors in several places. Consider careful revision with attention to grammar to improve readability.

Examples: Previously study showed that the Bi-OTA128 strain produced a high-molecular-weight EPS with a special O-acetyl group in its structure, which could inhibit inflammatory responses by decreasing NO, TNF-α, IL-1β, and IL-6 levels in lipopolysaccharide (LPS)-treated RAW264.7 cells (Figure S1). Line 82 onwards…

Reviewer 2 Report

Comments and Suggestions for Authors

In general, this is a well-planned and performed study, the analyses and presentation of the obtained results is clear and easy to follow. The title “A Ropy Exopolysaccharide-Producing Strain Bifidobacterium pseudocatenulatum Bi-OTA128 Alleviates DSS-Induced Colitis in Mice” and the way the Authors describe and interpret the results indicates that Bifidobacterium pseudocatenulatum Bi-OTA128 strain showed its beneficial effect when applied after DSS intervention and reduces intestinal damage, inflammation and microbiota shifts caused by DSS. Eg. Line 18 “The DSS-induced oxidative stress was reduced after B. pseudocatenulatum treatment…”. Line 264-265 “These results indicated that B. pseudocatenulatum intervention could alleviate DSS-induced colon damage and recover mucous layer…” and Line 496-499 “Notably, the EPS-producing strain Bi-OTA128 also reduced the relative abundance of Enterorhabuds, Halomonas, Romboutsia, Lactococcus, and Enterobacter which highly increased in the DSS-treated mice and recovered the gut microbiota composition to a balanced situation similar to the Control mice”

The manuscript, however, describes a protective effect of two Bifidobacterium strains (a strain-dependent effect) administration against DSS-induced colitis, which can be unambiguously read from the experimental design (Figure 1). Therefore, the manuscript should be corrected (re-written?) to avoid misunderstanding and overinterpretation of the observed effects. In the present form, it could be concluded that Bifidobacterium pseudocatenulatum Bi-OTA128 strain can be used as an agent supporting recovery from ongoing colitis, whereas the study was not intended to verify such a hypothesis.

The scientific community agrees, that every component or substance that is a subject of a study should be well characterized and the way it is obtained (isolated, extracted or purified) should be described in detail to enable other researchers to repeat the study. In this manuscript, the Authors refer to their previous results on the effect of EPS produced by B. pseudocatenulatum Bi-OTA128 on inflammatory parameters of LPS-induced RAW264.7 cells, and show them as supplementary information (Figure S1). These analyses are not described in the manuscript, there is no information on the process of EPS isolation and preparation, therefore, in my opinion, cannot be a part of the article in the present form. I would recommend that the Authors publish in the first place (before this article) the results of in vitro studies conducted on the EPS alone or refer to this data as “unpublished data”. The latter will prevent future reference to the results of non-peer-reviewed studies.

Minor remarks

Table S1 and S2 – please fill all the cells with the descriptive words of analyzed parameters

Figure 1 please delete “normal” from “normal saline”

The presentation (description in text) of changes in mRNA expression as percentages should be avoided as it is calculated as the relative value. It could be appropriate if the absolute gene expression was performed (and expressed as the copy number of specific mRNA).

Line 397 “Oscillibacter. However”

Reviewer 3 Report

Comments and Suggestions for Authors

The manuscript by Wang et al. aimed to study the role of Exopolysaccharide-Producing Strain (EPS) Bifidobacterium pseudocatenulatum Bi-OTA128 in modulating dextran sulfate sodium (DSS)-induced colitis mouse model. The overall results suggest improved intestinal barrier integrity, anti-oxidative stress, and anti-inflammatory effects of Bi-OTA128 in treating DSS-induced colitis. These findings greatly improve our understanding in this new EPS producing strain and will bring important insights and guidance for how to use this strain as a novel probiotic to benefit patients.

Here lists a few of my concerns that could improve the overall quality of the manuscript if addressed appropriately.

Q1: Introduction:

Line 57-78. Since the manuscript is studying the different treatment between EPS-producing strain and non-EPS-producing strain, the differences between these two strains needs to be illustrated and introduced here based on what have been reported in this area. Particularly, the background of BLYR01-7 needs to be introduced. More references are needed for this section.

Line 79-85. Reference needs to be cited to show how Bi-OTA128 was discovered and isolated, and what kind of research has been done on this new strain.

Q2: Method:

Line 100. The gender of the mice played a role in the study, the authors needs to explain why only male is being used for this study and how it may impact its results and conclusions.

Line 198-204. For comparing treatment group versus the control group, Dunnett’s test is more common, and should be considered. For comparing treatment differences among more than two groups, Tukey’s HSD is more powerful than the Student’s t-test which only compares two groups. The author needs to re-evaluate their statistical method before drawing any conclusions.

Q3: Results:

Figure 2. A/B: The DSS group was not marked any statistical significance sign compared to the Control group, this is the most important control across all the study and need to be addressed through the manuscript and all figures.

Figure 2. C/D: The DSS group should be marked with statistical significance sign if there are compared to the Control group, instead of labeling the Control group.

Figure 3. B: The DSS group was not marked any statistical significance sign compared to the Control group.

Figure 7. A/B/C: If the DSS group showed no difference than the Control group, this set of results are not convincing to draw any treatment effect, and thus can be removed to the supplementary information.

Q4: Discussion:

Consistent and inconsistent findings from the field needs to be reported and discussed here. Any limitations of the study should be addressed.

Comments on the Quality of English Language

Extreme long sentences need to be shortened.

Reviewer 4 Report

Comments and Suggestions for Authors

Dear Authors,

The article entitled ‘A Ropy Exopolysaccharide-Producing Strain Bifidobacte- 2 rium pseudocatenulatum Bi-OTA128 Alleviates DSS-In- 3 duced Colitis in Mice' addresses the anti-inflammatory effects of an exopolysaccharide (EPS)-producing strain Bifidobacterium pseudocatenulatum Bi-OTA128 through a dextran sulphate sodium (DSS)-induced colitis mouse model. Treatment with B. pseudocatenulatum improved DSS-induced colitis symptoms and preserved the integrity of the intestinal barrier by upregulating the expression of MUC2 and tight junctions. The results indicate that the anti-inflammatory effect of the EPS-producing strain B. pseudocatenulatum Bi-OTA128 on DSS-induced colitis may be considered as a probiotic with anti-inflammatory activity.

The work carried out is of very good quality and is supported by the data. The original article is very interesting but needs to be corrected before publication.

Below is a list of suggestions that I think would help to improve the manuscript.

Although the manuscript is well written and well organised, I would ask the authors to better describe the conclusions section.

1.         The figures are clear and well organised, but Figure 8 should be moved to the Discussion section and the comments connected with as well.

2.         Please describe the conclusion section in more detail.

3.         The English spelling and grammar of the manuscript should be checked

4.         Please check the style of the references

Best regards,

Comments on the Quality of English Language

The English spelling and grammar of the manuscript should be checked

Round 2

Reviewer 1 Report

Comments and Suggestions for Authors

Authors addressed my concerns.